



# **POET** (v0.1): Speedup of Many-Cores Parallel Reactive Transport Simulations with Fast DHT-Lookups

Marco De Lucia[1], Michael Kühn[1,2], Alexander Lindemann[3], Max Lübke[3], and Bettina Schnor[3]

[1]GFZ German Research Centre for Geosciences, Fluid Systems Modelling, Telegrafenberg, 14473 Potsdam, Germany
[2]University of Potsdam, Institute of Geosciences, Karl-Liebknecht-Str. 24–25, Potsdam-Golm, 14476, Germany
[3]University of Potsdam, Institute of Computer Science, Operating Systems and Distributed Systems, An der Bahn 2, 14476 Potsdam, Germany

**Correspondence:** M. De Lucia (delucia@gfz-potsdam.de)

**Abstract.** Coupled reactive transport simulations are extremely demanding in terms of required computational power, which hampers their application and leads to coarsened and oversimplified domains. The chemical sub-process represents the major bottleneck: its acceleration is an urgent challenge which gathers increasing interdisciplinary interest along with pressing requirements for subsurface utilization such as spent nuclear fuel storage, geothermal energy and $CO_2$ storage. In this context

we developed `POET` (`POtsdam rEactive Transport`), a research parallel reactive transport simulator integrating algorithmic improvements which decisively speedup coupled simulations. In particular, `POET` is designed with a master/worker architecture, which ensures computational efficiency on both multicore and cluster compute environments. `POET` does not rely on contiguous grid partitions for the parallelization of chemistry, but forms work packages composed of grid cells distant from each other. Such scattering prevents particularly expensive geochemical simulations, usually concentrated in the vicinity of

a reactive front, from generating load imbalance between the available CPUs, as it is often the case with classical partitions. Furthermore, `POET` leverages an original implementation of Distributed Hash Table (DHT) mechanism to cache the results of geochemical simulations for further reuse in subsequent time-steps during the coupled simulation. The caching is hence particularly advantageous for initially chemically homogeneous simulations and for smooth reaction fronts. We tune the rounding employed in the DHT on a 2D benchmark to validate the caching approach, and we evaluate the performance gain of `POET`'s

master/worker architecture and the DHT speedup on a 3D benchmark comprising around 650 k grid elements. The runtime for 200 coupling iterations, corresponding to 960 simulation days, reduced from about 24 h on 11 workers to 29 minutes on 719 workers. Activating the DHT reduces the runtime further to 2 h and 8 minutes respectively. Only with this kind of reduced hardware requirements and computational costs it is possible to realistically perform the large scale, long-term complex reactive transport simulations, as well as performing the uncertainty analyses required by pressing societal challenges connected

with subsurface utilization.





# 1 Introduction

The term *reactive transport* indicates coupled numerical models investigating interacting hydrodynamical, thermal and chemical processes of reactive fluids in porous or fractured media (Steefel et al., 2015). It finds wide application in subsurface
utilization such as: assessment of long-term safety of $CO_2$ storage or nuclear spent fuel repositories, understanding of ore formation and hydrothermal systems, or geothermal energy utilization. These models are severely challenging from a computational standpoint, forcing to set up geometrically coarsened, simplified domains with respect to those routinely tackled by purely hydrodynamical simulations (De Lucia et al., 2015), nonetheless requiring *many cores* infrastructures to achieve acceptable runtimes.

The chemical sub-process, although an inherently *embarassing parallel* task, represents the computational bottleneck in this class of coupled simulations. In practical real life use cases, an overwhelming 80 to 99 % of total CPU time is spent in the chemistry sub-process (De Lucia et al., 2015; Leal et al., 2020). For this reason, in recent years large efforts have been dedicated by the scientific community to the speedup of coupled reactive transport simulations. The focus has been on improving the numerical efficiency of the specialized chemical solvers, or on replacing chemistry altogether with statistical surrogates. To
our knowledge, only marginal attention has been put to optimal load balancing in the context of parallel computations.

In the next section we give an overview of recent related work concerning reactive transport simulators and acceleration strategies.

## 1.1 Parallel reactive transport simulators: State of the Art

The coupling between the involved processes of solutes transport within a fluid phase and their chemical reactions with rock-
forming minerals can be achieved in three ways (cfr. Steefel et al., 2015, and references therein also for the software cited below): on the one hand the *global implicit* approach, in which the geochemical subsystem is solved together with the PDEs representing the transport of solute species (implemented by, e.g., `PFLOTRAN`), and on the other the *operator splitting* approach, in which the processes are solved sequentially. The latter is furthermore subdivided in *sequential iterative* approach (SIA; implemented by, e.g., `HYTEC`, `TOUGHREACT`, `CRUNCHFLOW`) and *sequential non-iterative* (SNIA, `TOUGHREACT`,
`OpenGeoSys`). The sequential architecture is usually considered to be the most advantageous and flexible in terms of software development, since it scales much better with the number of considered chemical species and reactions, and it fully exploits the embarrassing parallelism of chemical simulations at each simulation time step. Furthermore, it allows using specialized simulators for each sub-process, whereas a global implicit scheme requires an ad-hoc implementation.

Traditionally, the reactive transport community has rather been focused on implementing parallelism of specialized sim-
ulators (e.g., Steefel et al., 2015; Beisman et al., 2015; Moortgat et al., 2020) in order to efficiently scale when computing on large HPC facilities. Since the parallelization of flow and transport is much harder than chemistry, these simulators rely on fixed domain partitioning, and the computational load of chemistry is addressed by using more CPUs. He et al. (2015) describes a parallelization strategy in which the available CPUs are pooled in two distinct groups, the first being used to solve flow and transport, and the second exclusively for geochemistry. In this scheme it is therefore possible to implement different





strategies for optimal load balancing of the geochemical sub-processes; however no further in-depth analysis for the achievable

performance gain is provided. Finally, advances in software engineering produced general-purpose multiphysics frameworks

able to solve arbitrary Partial Differential Equations (PDE) describing different coupled processes. These frameworks usually

offer features such as user-transparent parallelization on different hardwares and dynamic grid refinement (e.g., Permann et al.,

2020). Implementations of reactive transport simulators based on this kind of frameworks are being published (Damiani et al.,

2020; Kyas et al., 2020; Soulaine et al., 2021), but to our knowledge a detailed analysis of optimal parallelization specific to

geochemistry are not yet available.

A complementary research axis is represented by surrogate geochemical models, employed at runtime during the coupled

simulations in place of the expensive equation-based chemical solvers (Jatnieks et al., 2016; De Lucia et al., 2017). Surro-

gate models are statistical multivariate regressors which are trained in advance on an ensemble of geochemical simulations

spanning the expected parameter space encountered during the coupled simulations. Regressors commonly employed to this

goal are Artificial Neural Networks (Laloy and Jacques, 2019; Guérillot and Bruyelle, 2020; Prasianakis et al., 2020; Lu et al.,

2020), RandomForest (Lu et al., 2020), xgboost (De Lucia and Kühn, 2021), Gaussian Processes (Laloy and Jacques, 2019).

For low-dimensional problems - intending the number of free variables that define the behavior of a chemical system -, employ-

ing lookup tables proved to be a feasible approach (Huang et al., 2018; Stockmann et al., 2017). However, their applicability

degrades rapidly with dimensionality of the chemical system, since search and interpolation within the tables become exponen-

tially costly when dealing with many independent variables. All these approaches have in common the necessity to precalculate

the chemistry, and many research questions are still open concerning the required sampling density of the parameter space and

the optimal tuning strategy of the surrogates, which is a hardly automatable problem. Leal et al. (2020) suggest to store the

whole matrices representing already solved equilibrium states in lookup tables. A new equilibrium solution in the vicinity of

already stored points can be subsequently obtained by Taylor expansion, leveraging automatic differentiation implemented in

their geochemical solver. This greatly reduces the number of expensive equilibrium calculations, producing relevant speedups.

However, since all the components of the equilibrium geochemical system and their derivatives must be stored, further algorith-

mic optimizations involving clustering to reduce the time to identify the optimal neighbouring points for each new predictions

must be employed to avoid the curse of dimensionality (Leal et al., 2020; Kyas et al., 2020), potentially affecting the scalability

of the method.

## 1.2  Contributions of this paper

We initiated the development of POET in order to integrate and evaluate different algorithmic improvements capable to acceler-

ate coupled reactive transport simulations, in the framework of a SNIA coupling, and focussing in particular on the acceleration

of the geochemical sub-process. POET's parallelization of the geochemistry has been designed with a master/worker architec-

ture based on the MPI standard (Message Passing Interface Forum, 2015). Furthermore, we introduce a novel MPI-based

implementation of *Distributed Hash Tables (DHT)*, enabling caching of already computed geochemical simulations and their

subsequent reuse in further time steps. Since lookup and retrieval from the DHT is much quicker than the setup and calculation

of the corresponding "full physics" chemical simulations, the caching promises significant speedup when the already com-



puted values are frequently reused. This is a common occurrence in many practical reactive transport scenarios, which often
start from an homogeneous initial state and where a reactive front typically spreads through the domain with a high degree of
self-similarity (De Lucia et al., 2015).

Overall, the contributions of this work as implemented in `POET` and which are detailed in the paper can be summarized as:

– We propose a master/worker design and a load distribution algorithm to speedup geochemical sub-process (Section 2.2).

– We present a novel implementation of a fast MPI-based DHT (Section 2.4.1).

– We propose the concept of *approximated lookups*, where already simulated results are reused to approximate the results
of geochemical simulations with *similar* inputs combinations (Section 2.4.3).

– We validate the approximated lookup approach with respect to the variables' rounding (Section 4.1).

– We evaluate in detail the performance of `POET` with a 3D benchmark model comprising around 650 k grid elements and
200 coupling iterations, using up to 720 cores on a compute cluster, highlighting the linear scaling achieved with `POET`'s
architecture and the decisive benefit given by the DHT (Section 4.2).

## 2 `POET`'s Architecture and DHT Implementation

`POET` implements a SNIA coupling between the processes involved in reactive transport in porous media: hydrodynamic
flow, solute transport, and geochemistry (Figure 1). Since our focus lies on the runtime improvement of the geochemistry,
the current `POET` implementation of hydrodynamic flow and transport is quite simple. However, the generated results can
be directly transferred to many practical scenarios of subsurface utilization, which are extremely advection-dominated (high
Péclet numbers) and with kinetically-limited reactions, a common occurrence in, e.g., $CO_2$ storage scenarios (De Lucia et al.,
2015).

For historical reasons, `POET` is written in C++ wrapping large portions of high-level R code using the `RInside` frame-
work (Eddelbuettel et al., 2021). The C++ glue code is also responsible for the parallelization, implemented using MPI (Mes-
sage Passing Interface Forum, 2015). MPI is the de facto standard in the HPC community and supports fast point-to-point
communication and collective operations like broadcasts and barriers.

### 2.1 Flow and Transport in `POET`

`POET` uses at the moment of writing an advection with an explicit first order Euler scheme (forward time, with upwinding)
on irregular, unstructured finite volume discretizations of any shape and dimensionality. Some more details about this im-
plementation can be found in (De Lucia et al., 2017). The fluxes of the moving fluid phases across the interfaces of domain
elements are externally pre-computed using the multiphase simulator MUFITS (Afanasyev, 2013, 2015; De Lucia et al., 2016)
and loaded as "flow snapshots" at the beginning of the reactive transport simulations. Thus, there is no feedback, at the mo-
ment, between change in porosity following chemical reactions and hydrodynamics. Another important simplification is that,





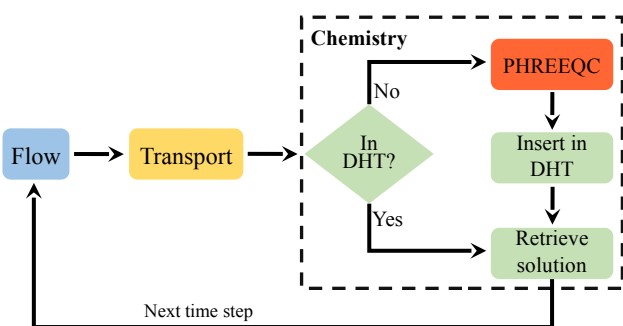

**Figure 1.** Schematic representation of Sequential Non-Iterative Approach (SNIA) coupling of the three sub-processes Flow, Transport and Chemistry. Distributed Hash Tables are filled at runtime with the results of "full-physics" chemical simulations, so that these can be reused at subsequent coupling iterations for *similar* chemical problems.

within a time-step of the coupled simulation, hydrodynamic flow is considered stationary. The time lags between available
successive flow snapshots define the time-stepping of the coupled simulations. For the internally computed advection, the
Courant-Friedrich-Levy condition for the maximum allowable time step of transport reads:

$$\Delta t_{adv} \leq \min_i \left\{ \left| \frac{V_i \cdot \varphi_i \cdot S_i \cdot \rho_i}{F_i} \right| \right\} \tag{1}$$

where the subscript $i$ refers to the $i$-th grid element, $F$ [kg s$^{-1}$] is the total mass flux of the transporting phase (water) across
the element boundaries, $S$ [/] the saturation of the water phase, $\rho$ [ton m$^{-3}$] its density, $V$ [m$^3$] the volume of the grid element
and $\varphi$ the porosity. If this $\Delta t_{adv}$ is less than the overall requested $\Delta t$, several "advective inner iterations" are computed before
calling chemistry. For this reason, the flow snapshots fed to the coupled simulation must be frequent enough, although no
requirement on constant time stepping is made. In the results presented in the remainder of the paper only a few time steps
required three or more inner advection iterations. This is a legitimate approximation for moderately transient flow regimes and
for moderate amounts of chemical reactions.
The geochemical reactions are computed using the established `PHREEQC` simulator (Appelo et al., 2013) through the R
interface (De Lucia and Kühn, 2013; De Lucia et al., 2017). Flow and transport are computed sequentially on a single CPU at
each iteration.

## 2.2  `POET`: Master/Worker Parallelization of Chemistry

The master/worker design is a coarse grain parallelization approach where the master process distributes *work packages* to the
worker processes. Each worker starts then the simulation with the received input parameters and reports the results back to
the master process. Since the communication is done via MPI, `POET` may run on multicore systems, but also on distributed
memory machines like compute clusters.

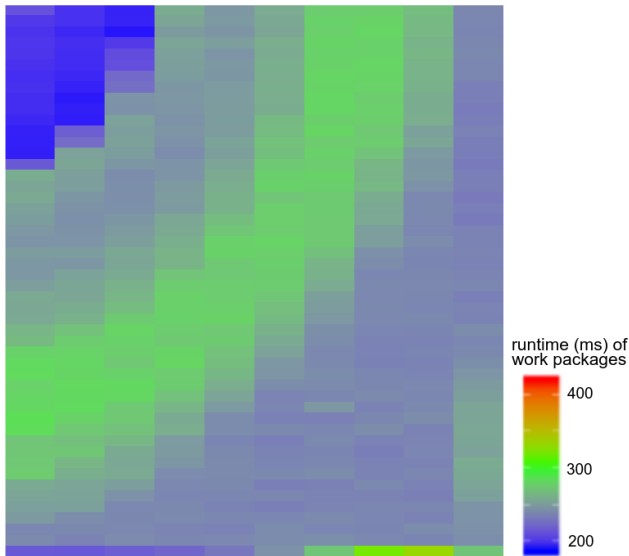

**Figure 2.** Runtimes for single work packages composed of 5 contiguous cells in an exemplary 2D 50x50 simulation. Chemistry is more computationally intensive near the reaction front, and contiguous domain partitioning can therefore suffer from suboptimal load balance.

In general, the simulation time of each single chemical computation depends heavily on the input parameters, and thus varies in space and time during the coupled simulations. While in some grid elements the chemical computation may only require few

iterations of the geochemical solver (elements either not yet reached by the reactive solution or where the bulk of the reactions has already happened), in the vicinity of the reaction front the simulation time is typically much higher. Figure 2 showcases the simulation time differences between blocks of five neighbouring grid cells for an exemplary iteration of a two-dimensional benchmark (described in Section 3.2). The reaction front propagates from the upper left corner to the middle of the field. While in blue regions the runtime is under 200 ms, the runtime in the green area of the reaction front is between 250-300 ms, which

is a significant 50 % increase. This shows that relevant work imbalances are detectable already for fairly simple scenarios: were all work packages comprising five cells run in parallel, the overall simulation time required by geochemistry would be controlled by the most computationally intensive one, located at the reaction front. In computer science terminology, this is called a heterogeneous workload.

In case of the heterogeneous workload typical for reactive transport simulations, a static workload distribution such as that

depicted in Figure 2, in which each worker is assigned a fixed number of grid elements, is particularly prone to unbalanced loads, where some workers are still computing while others are already idle. For POET , we chose instead a *dynamical* work load distribution, which decouples the number of workers and the dimension of the work packages. Furthermore, instead of composing work packages with contiguous domain partitions, POET packs a user-specified number of grid elements far from each other into the work packages, in a domain-independent algorithm often referred to as Round Robin (Figure 3, for a

scenario with 4 workers).

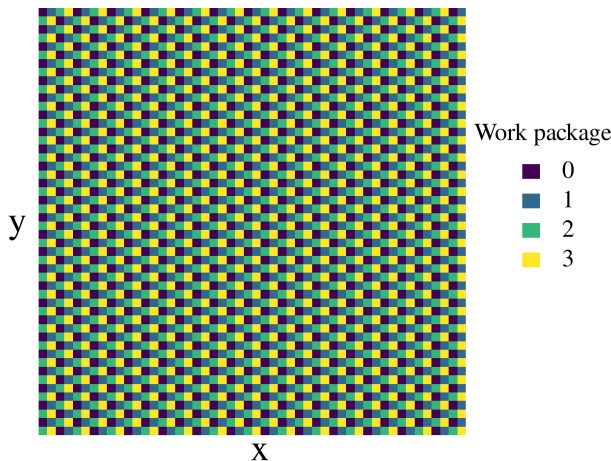

**Figure 3.** A Round-Robin distribution of grid elements into four work packages for the two-dimensional 50x50 grid of the `SimDol2D` scenario. Thereby, the elements with high compute demands, corresponding to the passing reaction front, are automatically distributed over the available workers.

With this design choice, the work for areas with high workload will automatically be well distributed across different work packages throughout the simulation run. Since we combine the Round-Robin partitioning with dynamic load balancing, the partitioning is only important for constructing the list of the work packages, and the maximal length of a work package is a parameter that the user can specify depending on the problem at hand and the compute cores available. The influence of this parameter is evaluated specifically in Section 4.2.3.

POET's master loop is shown in Figure 4. During the first coupled iteration, the domain elements are assigned to a given work package. This Round-Robin partitioning is static throughout the simulation, and the indices pointing from each domain element to its work package are stored by the master process. A coupling iteration starts by reading the Darcy velocities from the MUFITS snapshots, upon which solutes' advective transport is computed. Then the partitioning indices are reused by the master process to fill a two-dimensional row-major buffer in which all the domain elements belonging to a work package are contiguous. The chunks corresponding to a work package are then dispatched dynamically to the free workers. This means that in a first round, the master sends work packages to all idle workers and then waits for the results. When the reply message of a worker is received, the master distributes the next work package to the now idle worker. This is done as long as there is still work to do, i.e. as long as there are still grid elements to simulate. After all work packages have been processed, the master proceeds to reassemble the original list of domain elements with the updated concentrations, and after performing some post-iteration operations (such as writing the current snapshot onto disk), it advances to the next coupling iteration.

## 2.3 Distributed Hash Tables: related work

Distributed Hash Tables are frequently employed to enable fast distributed storage and data retrieval especially for applications in big data and data analytics. Prominent examples are Redis (Red, 2021) and Memcached (Mem, 2021) which are used by





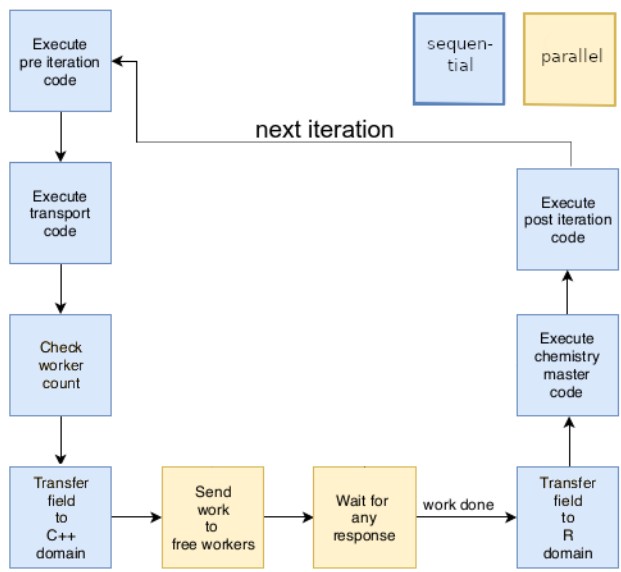

**Figure 4.** Overview of `POET`'s Master-Loop.

high-throughput web applications like Facebook and Netflix. Since these systems follow the client-server-architecture, the user would have to setup the DHT as an additional software component which runs separately. While systems like D1HT (Monnerat and Amorim, 2015) use their own communication framework, there exist several MPI-based designs (see for example Tsukamoto and Nakashima, 2010; Wozniak et al., 2010; Li et al., 2016; Maynard, 2011).

Processes within an MPI application are addressed uniquely via a `communicator` and a so-called `rank` within the `communicator`. Tsukamoto and Nakashima (2010) implement a DHT using an algorithm adapted from the Chord algorithm (Stoica et al., 2003). They focus on the dynamic communication behaviour of the algorithm and propose the MPI routines `MPI_Comm_accept` and `MPI_Comm_connect` which establish communication between sets of processes which do not share a `communicator`. This approach does not fit to our use case since all `POET` workers belong to the same MPI application and therefore share the `communicator` `MPI_COMM_WORLD`.

Wozniak et al. (2010) present an implementation called C-MPI of the Kademlia DHT (Maymounkov and Mazières, 2002) for grid and HPC environments. Their implementation is accessible through a put/get C-MPI API which is built on a custom event-driven, remote procedure call (RPC) library called MPI-RPC. MPI-RPC allows the programmer to register local functions for invocation by remote processes over MPI. This is again not suited for `POET`, since all `POET`workers are running within the same application.

Gerstenberger et al. (2014) have measured performance scaling of a distributed hash table up to 32k cores. They implemented their own MPI-3.0 RMA library for Aries and Cray Gemini interconnects named foMPI (fast one-sided MPI). Inserts are based on atomic compare and swap (CAS) and atomic `fetch_and_op` operations which are implemented on top of proprietary Cray-specific APIs.



Christgau and Schnor (2017) also motivate the use of MPI's one-sided communication API for the implementation of an DHT. They present an algorithm for the implementation of MPI's passive target synchronization with Readers and Writers semantic. The implementation is suited for non-cache-coherent many-core systems like Intel's experimental Single-Chip Cloud Computer (Howard et al., 2010).

`POET`'s DHT design is also based upon MPI's one-sided communication API. The benefit of the one-sided MPI API is that the basic operations, `MPI_put` and `MPI_get`, are typically implemented using fast Remote Direct Memory Access (RDMA) on networks like InfiniBand (Mellanox Technologies Inc., 2003).

### 2.4 The Design of `POET`'s MPI-based DHT

Before a worker starts a geochemical simulation, it checks whether a *similar* value has already been simulated and stored in the DHT. This section presents the design of `POET`'s DHT implementation.

We use the following naming conventions. The memory where a key-value pair is stored is called a *bucket*. The *index* is the offset of a bucket within the memory of the target process. The worker process which stores the key-value pair is called the *target* of the read/write operation.

The DHT shall fulfil the following requirements:

– **Distributed Memory:** The DHT shall support large parallel runs on compute clusters.

– **Usability:** `POET` shall be self-contained for easy use and the installation of additional cache servers like for example Memcached and Redis shall be avoided.

– **Readers & Writers semantic:** The parallel read and write operations of the workers have to be synchronized. Concurrent reading is allowed, but a write request has to be done exclusively on the data item.

– **Collision Handling:** Since the use of an hash function may lead to collisions, there has to be some form of collision handling implemented.

Since `POET` shall be easy to use as an parallel application without the installation of additional cache servers, we have implemented the hash table within the `POET` worker processes with the help of additional library calls. We distribute the hash table over all workers to support the scalability of the application. Having only a single worker serving as a data store may induce memory problems and the corresponding worker may become the bottleneck of the application. Hence, in `POET`'s architecture, the hash table is distributed over all worker. We need message passing support for the communication between the worker process which wants to read or write into the DHT and the worker process which is responsible for the corresponding DHT bucket. We use the high performance communication library MPI for the implementation of the DHT. Thus the DHT will benefit from any improvement made within the MPI library. For example, MPI implementations like MPICH, OpenMPI, MVPICH and vendor-specific MPI implementations like Intel's MPI exploit Remote Direct Memory Access (RDMA) offered by InfiniBand networks (Mellanox Technologies Inc., 2003).





### 2.4.1 `POET`'s DHT Implementation

A distributed hash table stores (key, value) pairs, and any participating node can *efficiently* retrieve the value associated with a given key. Typically, a hash function is used for the fast mapping of the key to the address of the value. `POET`'s DHT API and its design follows the approach presented by Christgau and Schnor (2017).

The first design issue concerns the choice of the MPI communication primitives. MPI offers both one- and two-sided communication. For decades, two-sided communication via `SEND/RECV` was the most popular building block for parallel applications. But in that model, a process has to call actively a `MPI_Recv`. This communication model is suited for regular communication, where for example *neighbouring* processes have to exchange ghostzones (Foster, 1995).

A DHT does not follow this model since the next communication depends on the look-up key and is not known in advance. An implementation making use of two-sided communication, based on `SEND` and `RECV` operations, would be cumbersome for such a dynamic communication pattern. Furthermore, compute time may be wasted in blocking `RECV`-operations. One-sided communication based on `PUT` and `GET` operations is a much better paradigm for a DHT implementation. It allows to specify the communication parameters by the *initiator* process only, i.e. the one that wants to read or write into the DHT, and does not require any interaction like a `RECV` call on the receiver/target side. Since the initiator process reads/writes the remote memory of the *target* process, these operations are also called MPI Remote Memory Access (RMA) operations (Message Passing Interface Forum, 2015). On networks like InfiniBand with RDMA support, these communication operations are implemented very efficiently without interaction with the remote CPU.

When the parallel MPI application starts, every process gets an unique identifier called `rank`. In `POET`'s master/worker design, the `rank` 0 process becomes the master, while all other processes are workers. Figure 5 shows the address space of the worker processes. All worker processes announce some part of their local address space for remote access. This is done in the following way:

1. First each worker allocates memory using `MPI_Alloc_mem`. This MPI function is recommended since it allows optimizations on shared memeory systems.

2. The memory is initialized with zeros.

3. The memory is announced to the other workers for remote access using `MPI_Win_create`.

Since the workers are running in parallel, the `put` and `get` accesses have to be synchronized. Here, a Readers & Writers model allowing reads to occur concurrently while inserts are done exclusively is advantageous. This coordination scheme is implemented using MPI's *passive* target synchronization (Message Passing Interface Forum, 2015) which provides *exclusive locks* for one writer and *shared locks* for many readers. Figure 6 shows the pseudocode where the `put` and `get` operations are protected by the corresponding `lock` calls.

In Appendix A are briefly described both the API (Application Programming Interface) and some useful additional features of `POET`'s DHT implementation.





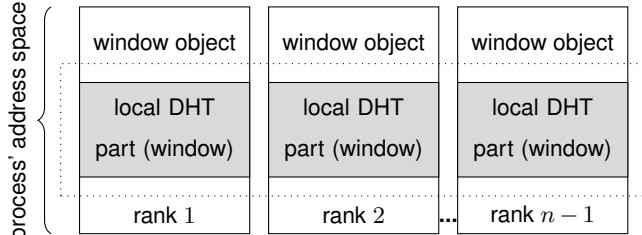

**Figure 5.** Usage of MPI windows for a distributed hash table.

| DHT_read | DHT_write |
|---|---|
| LOCK(w_obj, target, SHARED) | LOCK(w_obj, target, EXCLUSIVE) |
| GET(w_obj, target, &data) | PUT(w_obj, target, data) |
| UNLOCK(w_obj, target) | UNLOCK(w_obj, target) |

**Figure 6.** Readers &Writers semantic for concurrent accesses (Christgau and Schnor, 2017).

### 2.4.2 Addressing and Collision Handling

An MPI process is specified by its `rank` within a `communicator`. Hence, to localize the right bucket within the DHT, the `rank` of the bucket owner and the bucket index within the window is needed. The simulation input parameters are used as
260 key for the hash function. While the variables are stored as floats in double precision, they are rounded before hashing with a per-variable configured number of digits. More details about this aspect is given in section 2.4.3.

The default configuration takes md5sum as hash function, which delivers a 128 bit long hash value. To determine the rank of the responsible worker a simple modulo calculation is now used with the number of workers as modulus.

Indexing the bucket containing the actual data is a bit more complex. If we take the complete md5sum as index, this results
265 into $2^{128}$ buckets which describes a much too big index space. To address the complete DHT, each part must be at least $n$ bits in size so that $2^n$ will be greater or equal than the table size per process. Since the smallest arithmetic data type in C is a `char` and so a byte, $n$ must also be divisible by 8. Since the md5sum gives us more entrophy than needed for addressing, we use it to create an *index list* to store possible indices. The first free bucket is used for storing the value. In case that there are collisions for all indices, the key value pair in the bucket addressed by the last index is evicted. Analogously, if we lookup a table entry,
270 we test one index after another.

In our current implementation, the user defined hash function must return a 64 bit hash. Therefore, the md5sum hash is divided into two 64 bit parts, and both parts are bitwise XORed to deliver a 64 bit long hash. The 64 bits are again split into parts. The parts may also overlap each other. For example, in a setup with 7,017,920 buckets per process, the size of each index must be 3 bytes ($2^{24}$bits $\geq 7017920$) and the resulting index count is set to 6. The chosen indices are shown in Figure 7. Each
275 key value pair now has 6 possible locations in the hash table that can be used in case of collisions.





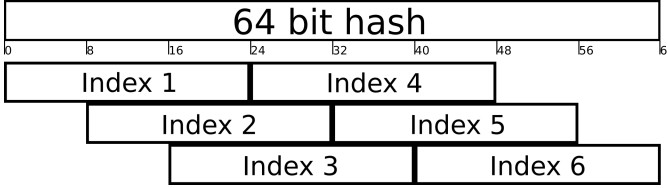

**Figure 7.** The hash value is used to determine the address of the corresponding bucket. Since collisions are possible, several possible addresses are derived from the hash value, and they are tested one after another.

### 2.4.3 Approximated Lookup

The DHT is employed in `POET` to store the results of geochemical simulations, which are largely multivariate both in input and output, by indexing them with a single key (the hash) from all the input variables. This means that a key-value data structure such as a hash table is only able to index an exact combination of input values. However, performing a rounding to user-defined significant digits for the input variables before computing the hash allows to relax such constraint and, in effects, defines the concept of *approximated lookup*, in which slightly different input combinations are in practice confounded. This in turn may introduce an error into the coupled simulations, which then propagates in space and time during the next iterations. It is therefore important to choose a rounding which maximises the reuse of already cached results while not causing significant errors. Finding the optimal rounding is a problem-dependent task and, given the nature of the numerical input values of geochemical simulations, which may vary of many orders of magnitude and which may carry different sensitivities on the results, further research is required to automate it.

`POET` lets the user decide how many significant digits are retained separately for each input variable, and also offer the capability to work with logarithmic variables. In the present version, if not differently specified by the user, the default rounding is at seven digits with logarithm.

### 2.4.4 Remarks

Since the DHT is integrated as a library within the MPI application and managed by the worker processes, no additional software installation is necessary. The DHT itself is built using MPI. Therefore, it is suited for both shared and distributed memory machines, and especially for compute clusters. Furthermore, we introduced the concept of an index list to efficiently handle collisions. In case all buckets referenced by the indices of the list are occupied, a simple eviction strategy overwriting the last bucket is used.

While MPI's concept of passive synchronization very comfortably supports the implementation of the Readers & Writers semantic, it puts additional synchronization overhead to the application. Furthermore, the synchronization is coarse-grained, since a writer locks the complete window instead of only locking the bucket where it wants to write. This synchronization overhead may eat away the benefit of the DHT-usage.





But before we go into the deeper performance evaluation in Section 4.2, we first validate whether the approximations associated with the DHT approach deliver trustful results.

## 3    Benchmarks definition

In the present work we first validate the approximated lookup using POET's DHT and secondly evaluate its performance. For the validation, which focuses on the choice of the rounding of inputs for the computation of the hashes and its influence on the
results w.r.t. reference simulations, a 2D domain of 2500 elements is employed. To evaluate POET's performance and parallel scaling on many-cores compute cluster, the same chemistry (and DHT parametrization) are solved on a much larger, 3D domain comprising around 650k elements.

### 3.1    Chemical Problem

The chemical problem solved throughout the present work derives from Engesgaard and Kipp (1992) and is commonly used,
with different variants, in the reactive transport community (e.g., Shao et al., 2009; De Lucia et al., 2017; Damiani et al., 2020; Leal et al., 2020; De Lucia and Kühn, 2021). A $MgCl_2$ solution enters a domain whose pore water is at thermodynamic equilibrium with calcite. The injected solution triggers the dissolution of calcite and the transient precipitation of dolomite. A total of eight variables thus completely define the chemical system. Six are related to aqueous concentrations and must be transported: the total elemental concentrations of C, Ca, Cl and Mg, plus pe and pH (transported respectively as $e^-$ and $H^+$).
The amounts of the two minerals, calcite and dolomite, are immobile.

The kinetics of calcite and disordered dolomite reactions are taken from Palandri and Kharaka (2004) removing the carbonate mechanism. The rate of precipitation of dolomite is set equal to the rate for dissolution. Furthermore, the reaction rates are considered independent of specific reactive surface area and thus from the quantity of mineral present in the grid element.

### 3.2    Validation benchmark: `SimDol2D`

For the purpose of validation of the DHT approach, the chemical problem defined in the previous paragraph is solved on a 2D, homogeneous cartesian square grid of 50 m side comprising 50x50=2500 elements (cfr. Figure 2). Porosity is constant at 0.25 and permeability at 100 mDa. The top left grid element is held at constant hydraulic pressure of 15 bar while the opposite side is held at constant pressure of 10. This results in a quasi-stationary flow regime, which is simulated with MUFITS for a total of 200 days, retaining one snapshot of flow velocities each 10 days, for a total of 20 snapshots. The coupled simulations are
however prolonged to reach 300 coupling iterations reusing the last available flow snapshot, to ensure that the whole simulation grid is reached by the reactive front by the end of simulation time. In the following we refer to this simulation scenario with the name of SimDol2D.





### 3.3 Performance Evaluation Benchmark: `SimDolKtz`

A much more challenging hydrodynamic setting called `SimDolKtz` has been used for the evaluation. This scenario depicts a

monophasic injection of a $MgCl_2$ solution in the same simulation grid used for the history-matched Ketzin pilot site for $CO_2$ storage (Martens et al., 2013; Kempka et al., 2013, 2014) near Potsdam, Germany, over a period of 60 days. The discretized domain contains 648,420 cells of different volumes and has spatially heterogenous porosity and permeability. The injection happens at constant rate in correspondence with the injection borehole of the Ketzin pilot site. Constant pressure boundary conditions are achieved setting pore volumes multiplicators at the outermost boundary elements of the domain. In total, 20

snapshots of the resulting MUFITS flow simulations are employed in the coupled simulations, the first 10 at time intervals of 1 day and the following 10 at 5 days lag, for a total of 60 days (20 iterations). Since for this case the time stepping is not constant, the $\Delta$t for the chemical calculations must be stored as additional input value in the DHT. The last flow snapshot is subsequently reused for 180 further coupling iterations, with the same 5 day lag, thus reaching 960 days in simulation time and 200 iterations in total.

### 3.4 Error Measure

The chosen synthetic measure for the comparison of the simulations with DHT w.r.t. the reference in a multivariate setting requires the normalization and combination of the different variables. We chose a relative root mean squared error using the maximum value of each variable as norm, combined with the geometric mean resulting in the expression of equation 2:

$$\text{Error}_t = \exp\left\{\frac{1}{m}\sum_i^m \log\left\{\frac{\sqrt{\frac{\sum_j^n (\text{ref}_{i,j} - \text{DHT}_{i,j})^2}{n}}}{\max_t(\text{ref}_i)}\right\}\right\} \tag{2}$$

where $m$ is the number of distinct variables to compare, $n$ the number of considered grid elements and $t$ the particular time step where the error is computed. A single numeric value, averaged throughout the domain, comparing multiple variables at once is of course susceptible to overlook discrepancies which, given the highly non-linear nature of chemistry, may in turn originate diverging trajectories in space and time. It remains hence very difficult to define an *a priori* threshold after which the approximated simulations must be rejected.

For the chemical problem considered in these benchmarks, a value of roughly $10^{-5}$ emerges *a posteriori* as a sensible limit. We would like to stress the fact that the same workflow we present in this work should be employed when using POET : some preliminary simulations on coarser or smaller grids should be run beforehand to check the retained digits for DHT and their influence on the results. Once optimal values emerge, as well as the confidence in the ability of a defined error measure to capture drifts or errors, then the larger, expensive runs should be launched.

## 4 Results



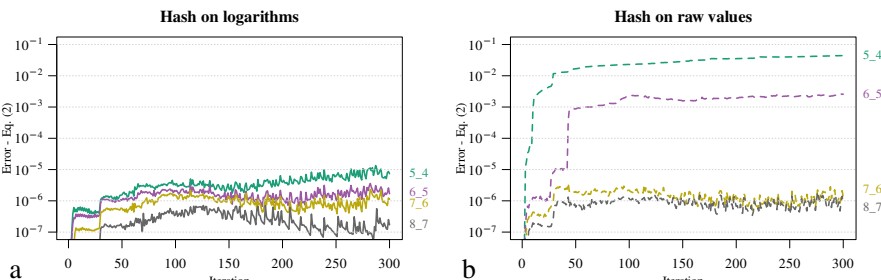

**Figure 8.** Error as defined by equation 2 along the simulations with DHT compared to the reference, as function of the retained digits for hashing. Colors and labels on the right margin encode the digits used for class 1 and class 2 variables respectively. **(a)** logarithm of variables used for hashing; **(b)** hash taken on raw variables.

## 4.1 Validation of Approximated Look-Ups

The chemical problem tackled in `SimDol2D` is expressed in terms of eight input/output variables. Our DHT implementation allows a granular, *per-variable* choice of digits retained for the hashing, to accomodate the possibility of larger precision for specific variables. In case of geochemical problems, it is sensible to assume, for example, that aqueous concentrations, including pe and pH, generally require a larger precision than minerals amounts. However, for simplicity, we divided these eight variables in two distinct classes, for which a different number of significant digits are used in the hashing algorithm. Hence we considered "class 1" the variables representing aqueous concentrations and "class 2" the mineral phases. We tested different numbers of significant digits for both classes: {5, 6, 7, 8 } for class 1 and {4, 5, 6, 7} for class 2. The cross-product of all these combinations were repeated using both the original variables' values for the hashing and their logarithms, for a total of 32 runs beside the reference.

Figure 8 shows the error as defined in equation 2 of four DHT simulations using different roundings for up to 300 iterations. Colors and labels on the right margin encode the digits used for class 1 and class 2 variables respectively. The left image shows the error when hashing logarithmic variables, and the right one when using raw variables. It is evident that taking the logarithms of the variables before hashing dramatically increases the accuracy of the coupled simulations with DHT if compared to the hashes taken on the raw variables with the same digits. Furthermore, all of the considered logarithmic simulations are in an acceptable discrepancy w.r.t. the reference, whereas this is true only for larger number of digits in the raw variables.

The errors introduced in the coupled simulations by approximated lookup can propagate in space and time and possibly lead to unphysical simulation results. The maximum deviations w.r.t. the reference of each DHT run across all iterations is therefore given in Table 1. The table is ordered starting with the entry for the largest digits (rounding 5_4) up to the smallest (rounding 8_7).

From this table it is clear that there is no unique order relation between the maximum errors and the digits used for the variable hashing. This means, for example, that the error for the logarithmic "5_7" case is larger than in the logarithmic "5_5" case. This can be interpreted as a consequence of the unreducible randomness in how the DHT is filled at runtime





and the actual magnitude of the error; even if the overall error is small, it is possible that a significant discrepancy of one
single variable, in certain regions of the parameter space, originates quickly diverging trajectories for the coupled simulations.
However, in general, logarithmic variables always produce acceptable to negligeable errors for this scenario, even with the
lowest number of considered digits: only in two cases their errors slightly trespass $10^{-5}$, which can be taken as a rough
threshold for satisfying accuracy of the end results. These cases are encoded in blue in Table 1. Again, much larger errors are
observed if the raw variables are used for hashing instead of the logarithmic values. Here, all simulations with 5 digits or less
for one of the two classes produced completely unacceptable results (encoded in red in the table).

**Table 1.** Validation of the `SimDol2D` scenario with DHT: maximum errors across all iterations for different retained digits. The logarithmic
case shows a clear advantage compared to the raw variables, being consistently more accurate. For this use case the total error should not
exceed roughly $1 \cdot 10^{-5}$. The unacceptable errors (leading to unphysical simulation results) are encoded in red; blue is used for acceptable
simulations and black are used for simulations with negligible accuracy losses.

| Class 1 | Class 2 | Log | Raw |
|---------|---------|-----|-----|
| 5 | 4 | $1.35 \cdot 10^{-5}$ | $4.41 \cdot 10^{-2}$ |
| 5 | 5 | $1.16 \cdot 10^{-5}$ | $5.27 \cdot 10^{-3}$ |
| 5 | 6 | $9.60 \cdot 10^{-6}$ | $3.43 \cdot 10^{-4}$ |
| 5 | 7 | $1.32 \cdot 10^{-5}$ | $5.25 \cdot 10^{-5}$ |
| 6 | 4 | $4.27 \cdot 10^{-6}$ | $3.19 \cdot 10^{-2}$ |
| 6 | 5 | $3.59 \cdot 10^{-6}$ | $2.62 \cdot 10^{-3}$ |
| 6 | 6 | $3.89 \cdot 10^{-6}$ | $7.65 \cdot 10^{-5}$ |
| 6 | 7 | $4.11 \cdot 10^{-6}$ | $8.01 \cdot 10^{-6}$ |
| 7 | 4 | $2.65 \cdot 10^{-6}$ | $3.15 \cdot 10^{-2}$ |
| 7 | 5 | $2.64 \cdot 10^{-6}$ | $2.06 \cdot 10^{-3}$ |
| 7 | 6 | $2.33 \cdot 10^{-6}$ | $3.70 \cdot 10^{-6}$ |
| 7 | 7 | $2.28 \cdot 10^{-6}$ | $3.63 \cdot 10^{-6}$ |
| 8 | 4 | $7.90 \cdot 10^{-7}$ | $3.15 \cdot 10^{-2}$ |
| 8 | 5 | $8.24 \cdot 10^{-7}$ | $1.77 \cdot 10^{-3}$ |
| 8 | 6 | $9.03 \cdot 10^{-7}$ | $1.84 \cdot 10^{-6}$ |
| 8 | 7 | $7.11 \cdot 10^{-7}$ | $1.97 \cdot 10^{-6}$ |

A visualization of the accuracy of the DHT results is given in Figures 9 and 10. They depict the scatter plots of the relevant
variables between the DHT coupled and the reference simulation in logarithmic scale after 300 iterations. Cl and calcite are
omitted from these plots since at the end of the simulations they are almost completely constant. We show the **raw 6_6** and
**log 5_5** cases respectively, which are both marked in blue in Table 1. These are cases where the inaccuracy of the end results
is acceptable, but already significant; it may quickly diverge after more iterations than the 300 considered in our scenario.

For the raw hashing with 6 digits for both variable classes in Figure 9, it's only the small values of C and Ca which depart
from the corresponding values in the reference simulations. However such discrepancy is only appreciable for values lower



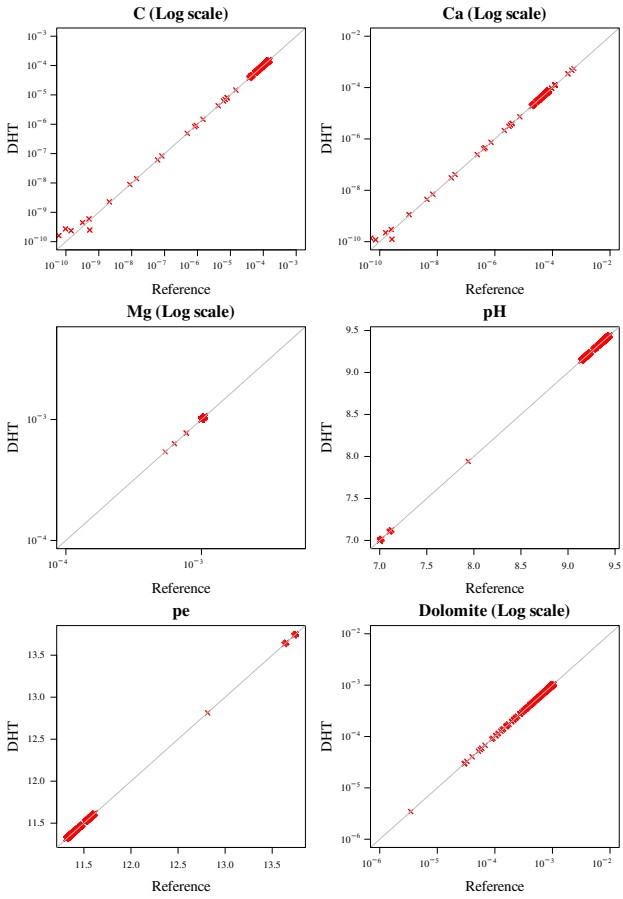

**Figure 9. Raw variables, 6_6 digits**: Scatterplot of DHT and reference variables after 300 iterations. While results of the DHT simulations are accurate enough, discrepancies are evident for small values of the variables.

than $10^{-9}$, whose "physical meaning" can be already assimilated to 0. The scatter plot of Figure 10, for the case of hashing of the first 5 digits of logarithmic variables, shows no inaccuracy even for the smallest values of C and Ca, down to $10^{-10}$.

Based on this preliminary study, the DHT with logarithm and 5 significant digits can be retained as accurate enough for the given chemistry.

## 4.2 Performance Evaluation

In this section we evaluate both the speedup achieved through POET's master/worker design with non-contiguous domain partitions and the further speedup achieved when employing the DHT. The Flow and Transport sub-processes are within the

sequential loop of the master and contribute significantly to the overall runtime (see Figure 1). Since our focus was to improve the dominating runtime of the Chemistry sub-process, we evaluated and report only the runtime of the latter, shortly named

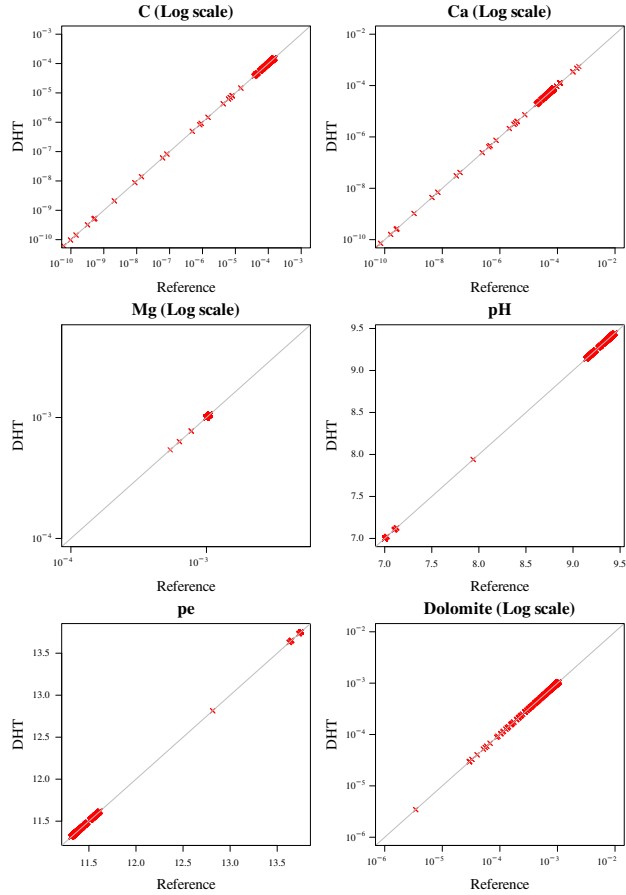

**Figure 10. Logarithmic variables, 5_5 digits**: Scatterplot of DHT and reference variables after 300 iterations. With only five digits for the logarithmic variables, the DHT simulations maintain excellent accuracy also for small variables values, as opposed to the case displayed in Figure 9.

*runtime* in the following. This avoids any influence of the other sub-processes. So, the time measurements were taken before and after the data transfer between the R and C++ domain (including all steps in the bottom row in Figure 4).

### 4.2.1 Testbed and Design of Experiments

All measurements described below were done on the compute cluster "Turing" of the Institute of Computer Science at the University of Potsdam. The cluster consists of 30 compute nodes, each equiped with two Intel Xeon E5-2650v4 with 12 cores and 64 GiB of main memory accessible for both CPUs. The compute nodes are connected to each other via one InfiniBand switch. An overview of the hardware and software configuration of the compute cluster Turing is given in Table 2.

Table 3 shows the design of our experiment. For evaluating the scalability of our parallelization we varied the number of
workers from 11 up to 719. The runtime without DHT using 11 workers is about 25 hours. Therefore, each experiment was





**Table 2.** Hardware and software specifications for the HPC test environment "Turing".

| Specification | Value |
|---|---|
| CPU | 2x Intel Xeon E5-2650v4 (12 Cores, 2.20 GHz) |
| RAM | 64 GiB |
| Network | InfiniBand |
| OS | Debian 9 |
| Compiler | GCC 7.2.0 |
| MPI-Framework | OpenMPI/4.0.0 |
| R-Runtime | 3.3.3 |

**Table 3.** Summary of investigated factors for the performance evaluation of `POET` in the `SimDolKtz` benchmark.

| Factor | Values |
|---|---|
| Use of DHT | on, off |
| Workpackage size | 4, 16, 128 |
| Count of worker | 11, 23, 47, 95, 191, 383, 575, 719 |
| Size of DHT | 1 GiB per worker |
| DHT: Rounding | 5 (all variables) |
| DHT: log variables | yes |
| Repetitions | 3 |

repeated only three times and the median is reported. Since the variation of the measurements was neglectable, three repetitions seem to be sufficient for trustworthy conclusions.

In all measurements, the master was running on a dedicated core. In the first experiment only half of the number of cores of one node was used, then all cores of one node (dense mapping), and in the following the number of nodes is doubled until the application is running on all cores of all 30 cluster nodes.

Since we were interested in the benefit of the DHT, we performed the experiment with and without DHT activated. The DHT configuration was the one which has proven to be accurate enough for the given chemistry (see Section 4.1). Further, we investigated the influence of different work-package sizes. The varied factors with their values, and further settings are summarized in Table 3.

### 4.2.2 Speedup and Scaling

Figure 11 shows the median of the runtime in seconds for different worker counts. Both axes are in log-scale. `POET` shows indeed a good linear scaling: the master/worker parallelization performs very well. The runtime is reduced from about 23 h on 11 workers to 29 minutes on 719 worker (work-package size 128). Furthermore, the benefit of the DHT is clearly visible. The





runtime for 11 workers only by activating the DHT is reduced to about 2 h 5 minutes. For the biggest run where the complete

cluster is used, the runtime is reduced to about 8 minutes with DHT.

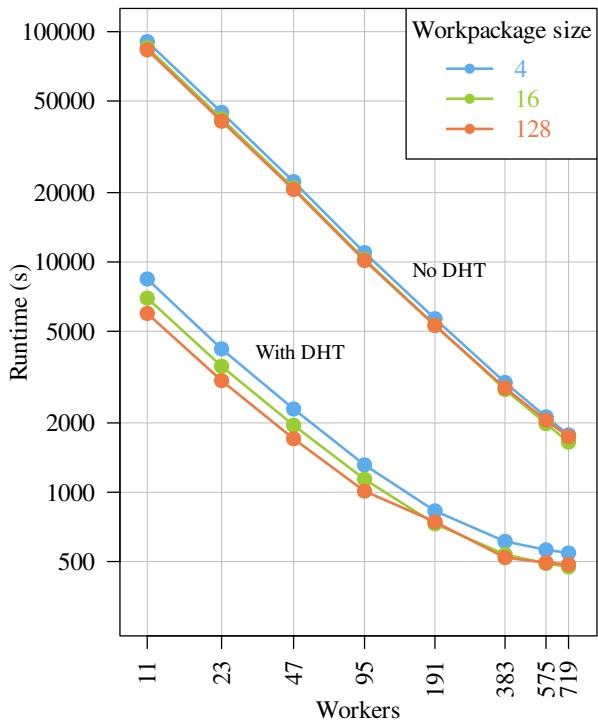

**Figure 11.** `POET`'s runtime with and without DHT activated and for different work package sizes (both axes in log-scale).

The runs with DHT are much faster, but do not scale as good as the runs without DHT. In Figure 12 we zoom into the runtime results with activated DHT. In this figure only the x-axis is in log-scale. It can be seen that there is only little improvement when using more than 191 workers. Figure 13 shows the corresponding speedup compared to our base measurement with 11 workers: $S(n) = \frac{t(11)}{t(n)}$, calculated for each work-package size. Since we approximately double the number of workers on

the x-axis, the expected ideal speedup also doubles. Figure 12 confirms this expectation for 23 worker and for 47 worker where the speedup is about 2 and 3.6, respectively. For higher worker numbers, the speedup diverges from the ideal speedup.

The speedup for the work-package size 128 is lower than for size 4 and 16, but this derives from the fact that the run with work-package size 128 on 11 workers is much faster than with the other work-package sizes, and therefore there is not much space for improvement/speeup anymore. All three speedup curves in Figure 13 have in common that the speedup converges

towards a constant. The reason for this behavior may be an increasing parallelization overhead compared to the simulation grid size or a non-neglectable sequential fraction. For example, when the number of worker increases, the synchronization overhead within the DHT library increases. But there are lots of other factors which have an influence on the performance. To investigate this in more detail, we calculated the corresponding Karp-Flatt metric (Karp and Flatt, 1990):



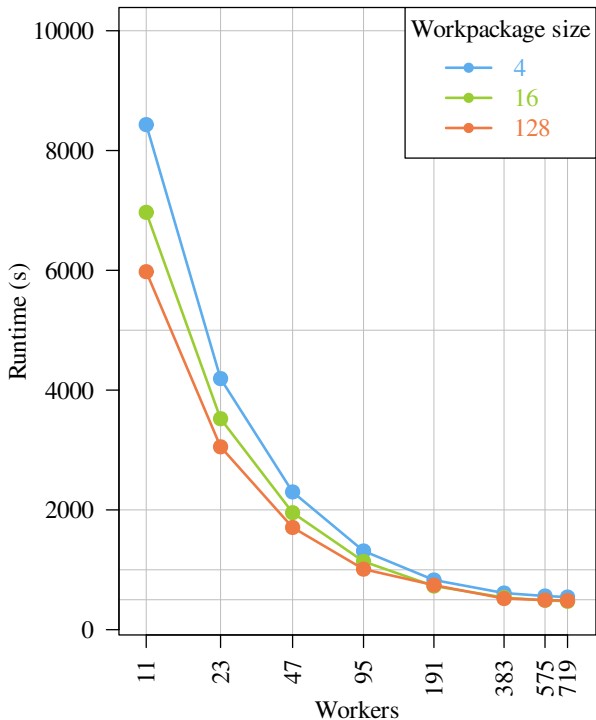

**Figure 12.** Runtime of `POET` with activated DHT (x-axis in log-scale).

$$f = \frac{\frac{1}{S(n)} - \frac{1}{n}}{1 - \frac{1}{n}} \tag{3}$$

In Table 4 we show the Karp-Flatt metric for work-package size 16. Since the Karp-Flatt metric is getting constant, this indicates that there is a sequential component within the application which limits further scaling. For work-package size 16, this fraction is equal to 7%. With a sequential fraction $f = 0.07$, we deliver an asymptotic speedup of $\lim_{n\to\infty} S(n) = \frac{1}{f} = 14.3$ from Amdahl's Law. This explains the course of the speedup curves.

To confirm this observation, we instrumented the `POET` code with additional time measurements. The master spends each iteration about 2 seconds in the data transformation from the R to the C++ domain and vice versa. So, after 200 iterations about 400 s of the runtime is contributed by this data transformation. Starting with 11 workers, the runtime with DHT activated was about 2h 5 minutes. Therefore, about 5.3 % of the sequential fraction comes from the data transformation.

### 4.2.3 Influence of Work-Package Size

The work-package size has a clear influence on the runtime. This influence decreases with a higher degree of parallelism, i.e., a larger number of workers. For all three tested work-package sizes, we observed a similar good asymptotic runtime.



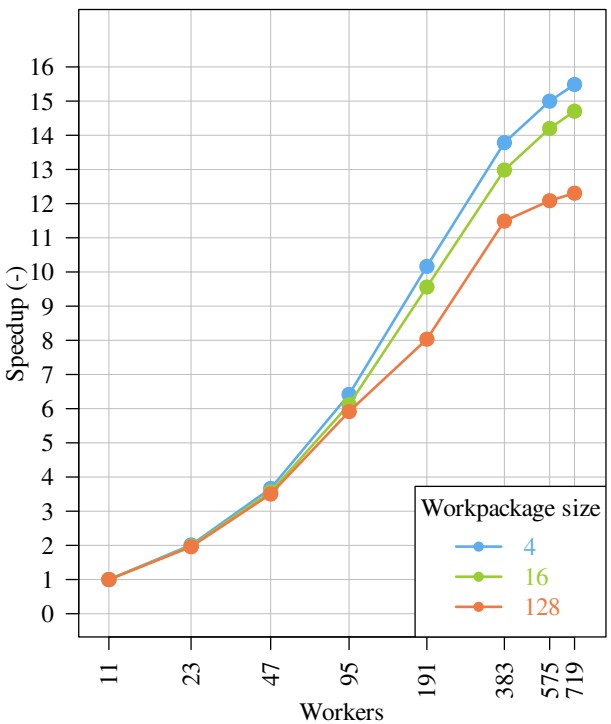

**Figure 13.** Speedup with respect to the reference simulation with 11 workers and active DHT.

**Table 4.** Karp-Flatt metric with activated DHT and work package size 16.

| Workers | Karp-Flatt metric |
|---:|:---:|
| 23 | 0.48 |
| 47 | 0.26 |
| 95 | 0.15 |
| 191 | 0.10 |
| 383 | 0.07 |
| 575 | 0.07 |
| 719 | 0.07 |

There is some overhead associated with the function call of the chemical solver, due to the need for composing a string buffer expressing the required computations in `PHREEQC`'s syntax; this overhead may have a non-negligible influence. For example, for a node of the Turing cluster and our benchmark configuration, we approximated the overhead by the following formula:

$$y = 0.00568 + 0.00703 \cdot x \ [s] \tag{4}$$





where $x$ gives the size of the work-package. We conclude that for the chemistry of our benchmark the `PHREEQC` simulation time for one cell is only 24 % higher than the base overhead due to the function call. In case of 4 cells per work-package, 162,105 calls per iteration will occur $(648420/4)$, without DHT usage. In case of 128 cells within a work-package, this number is reduced to only 5066 `PHREEQC` calls. This explains the better performance of the runs with larger work-package size. With an increasing number of workers more calls are done in parallel, and this influence tends to vanish.

Regarding an optimal work-package size, our investigations are just at the beginning and in future work it would be necessary to develop support for an automatic setting of a *suited* work-package size, possibly adjusted dynamically during the coupled simulations.

### 4.2.4 DHT Usage and Filling

Each data point is first looked up in the DHT to determine if there is already a previously simulated result. So there are always
$200 * 648,420 = 129,684,000$ calls to `DHT_Read()`. Table 5 gives the hit rates after 200 iterations. These are obtained by dividing the number of hits by the number of all DHT reads. The results of all three runs were averaged using the arithmetic mean. As expected, the influence of the number of workers and the different work-package size on the hit ratio is minimal. It is evident that for the `SimDolKtz` benchmark an enormous amount of about 94.4% of the look-ups are successful.

**Table 5.** Hit rate after 200 iterations on *SimDolKtz*.

| | Hit-Rate | workpackage size | | |
|---|---|---|---|---|
| | | 4 | 16 | 128 |
| | 11 | 94.46% | 94.46% | 94.47% |
| | 23 | 94.48% | 94.46% | 94.45% |
| worker | 47 | 94.45% | 94.48% | 94.44% |
| | 95 | 94.46% | 94.48% | 94.45% |
| | 191 | 94.45% | 94.46% | 94.41% |
| | 383 | 94.42% | 94.45% | 94.37% |
| | 575 | 94.45% | 94.43% | 94.38% |
| | 719 | 94.41% | 94.46% | 94.32% |

Furthermore, we can determine the fill level of the DHT for `SimDolKtz`. Key and value consist each of eight input variables
and additionally stored time step (see Section 3.3). All 9 parameters are stored as double values. Furthernire, there are 72 bits management data per bucket. This results in $9 * 2 * 64\,\text{bits} + 72\,\text{bits} = 1224\,\text{bits}$ per bucket. From Table 5, we can conclude that about $(129,684,000 * (1 - 0.944)) * 1224\,\text{Bits} \approx 1,035\,\text{GiB}$ of memory is occupied over the entire runtime. It follows that even





for the smallest configuration with 11 workers, the DHT is filled only to $\frac{1.035\text{GiB}}{11\text{GiB}} \approx 9.4\%$. As the number of workers increases, the total memory available to the DHT will too, so that the fill level will further decrease.

Finally, a note on the number of evictions. Due to the experiment design, we did $3 \times 8 \times 3 = 72$ measurements with activated DHT. Only during five of these 72 measurements an eviction occurred. The maximum number of observed evictions during one run was three. Hence, the DHT size is an important tuning parameter to reduce POET's memory demand and should be adapted fitting to the scenario which the user wants to simulate.

## 5 Discussion

The scientific simulator POET has been developed to serve as testbed for novel ideas aimed at increasing the computational efficiency of parallel reactive transport simulations, with specific focus on the geochemistry sub-process. This project can be further improved both from the standpoint of technical implementation and of the offered capabilities. In particular, the blend of high-level language R and more efficient but more implementation-heavy C++, while offering a high flexibility for quick algorithmic prototyping, shall be progressively refined in favour of the latter, hence removing significant bottlenecks. Never-

theless, it is already possible to highlight some remarkable results from the algorithms and numerical experiments presented in this work. In our best knowledge, POET represents the first implementation of dynamical load distribution architecture via master/worker model specifically for reactive transport simulations, whereas a similar concept had already been discussed by He et al. (2015).

Different parameters pertaining POET's parallelization and DHT are left for the user to set in absence of a reliable way
of determining their optimum at runtime. We refer here in particular to the work-package size and to the rounding applied to each input variable before hashing. Specifically to the rounding, which is an extremely problem-dependent parameter, we recommend a workflow in which different DHT roundings are tested on smaller (and hence quickly computed) domains in order to assess their influence on the results, and only start large runs once the least digits not leading to significant discrepancies w.r.t. reference (i.e., simulations without DHT) are identified. A conservative default setting is used by POET if the user does
not specify otherwise, but it may be suboptimal, leading to unnecessary filling of the table and larger runtimes.

We stress the fact that the caching approach is completely problem-agnostic and hence applicable in any reactive transport code, and further in any coupled simulator where sub-processes are expressed in terms of input/output variables. The available RAM is the only limiting factor. Enabling DHT partially transforms a computationally intensive problem into a memory intensive one, where each worker process reserves some RAM devoted to the local DHT portion. At the moment, POET's DHT
implementation has no advanced memory management: once the reserved RAM has been filled with simulation data, the DHT acts as LIFO (Last In, First Out) buffer to accomodate new records. This aspect needs further implementation work in order to enable either a FIFO (First In, First Out) approach or preferential discarding of records following a usage metric. However, the SimDolKtz test case was run with only 1 GiB RAM assigned to each worker for the local DHT, which is a quite small amount for recent workstations or for cluster nodes, and even in the most unfavourable case (with 11 workers plus the master
process) the DHT was filled at most to 10 %. This of course can vary with more complex chemistry requiring storage of more





variables and most of all with many more coupling iterations. A considerable advantage of the DHT approach is that query and retrieval of data will scale very well also when storing many more variables than those used in this work, a benefit due to the key-value format of the table. Hence, the speedup achievable with DHT caching will be even more pronounced for more complex chemistry.

In current `POET`'s version we chose a rather simplistic approach for the coupling of chemistry, in which all "state variables" defining completely the chemical problem (aqueous concentrations and mineral amounts) and the corresponding outputs are saved. It is possible to spare DHT space by recasting the results of the chemical problem in terms of the actually independent variables (i.e., storing reaction rates within the time step along with pH and pe and the $\Delta t$, and to back-calculate the changes in total elemental concentrations (De Lucia and Kühn, 2021), possibly in combination with scaling relationships such as those

proposed by Klein et al. (2013) and De Lucia et al. (2015), which could also be beneficial in case of spatial heterogeneity not only of porosity and water saturation, but for the chemical process itself. This is however a quite complex task, since the particular definition of kinetic laws and the choice of the relationships used to update parameters such as specific reactive surfaces, which we did not consider in the present work, directly affect how the "chemical heterogeneity" can be treated to optimize the speedup through a DHT. An advance in this regard would be particularly interesting since it is trivial to understand

that the DHT is particularly suitable for simulations starting from a chemically homogeneous initial state and where smooth reactive fronts propagate within the domain. In this sense, the DHT includes and generalizes to parallel environments the compression algorithm devised by De Lucia and Kühn (2013) with the added benefit of including all previous iterations in the "compression". The initially homogeneous case is in itself a simplification commonly observed in the praxis of reactive transport, and is determined on one hand by the lack of knowledge about the underground, which can only be attacked by Monte

Carlo studies, and on the other by the large computational cost associated to each coupled simulation, which makes those Monte Carlo studies unfeasible. `POET`, aiming at mitigating precisely these aspects, can serve as basis for further developments and research. In particular, the DHT approach is flexible and can be easily adapted to any particular problem and strategy, i.e. also storing partial results such as species concentrations, partition coefficients or mineral saturation indices, if explicitly required by the chemical problem in the simulations, with relatively simple additional programming. Furthermore, the dump

and restore capability already implemented in `POET`'s DHT allows to start new simulations with a DHT already filled by previous runs, potentially saving enormous runtime if for example testing the effects of changing boundary conditions or permeability distributions, given the corresponding MUFITS simulations are available.

The order of execution of each work-package in `POET` is inherently non-deterministic, and this fact leads to slightly different speedups, hit ratios and results observed in repeated runs when running with DHT active, since also the DHT is non-

deterministically filled. This issue is unavoidable, due to the dispatch of work-packages in a first come, first serve basis; it is however very moderate, and the advantage given by dynamic load distribution outweighs it by ensuring an efficient hardware usage. With appropriate rounding, the results of the simulations with DHT display insignificant differences w.r.t. the reference and can thus be fully relied upon.

Overall, with enabled DHT we were able to run 200 coupling iterations on a 650 k elements domain (`SimDolKtz` test case)

in under 6000 s using 11 workers. The corresponding reference simulation with no DHT took almost 83000 s, more than 14



times slower. This speedup puts the use of DHT in the same tier with other acceleration strategies such as surrogate modelling (Jatnieks et al., 2016; Laloy and Jacques, 2019; Prasianakis et al., 2020; De Lucia and Kühn, 2021) and the *on demand machine learning* approach of REAKTORO (Leal et al., 2020; Kyas et al., 2020). In particular, the caching via DHT is an algorithm which, once implemented in the coupled simulator and tuned for the problem at hand, does not require training of complex

surrogates but it works out-of-the-box. Furthermore, it scales well for both massively-parallel simulations and even better for more complex chemistry than the ODML. Nevertheless, it should be regarded as one tool in the arsenal of the scientist and not as an alternative.

In the next development steps, along with the above mentioned technical improvements resulting from the reimplementation in native C++ of some R routines, we envisage notably the adoption of geochemical surrogate models for chemistry (De Lucia

et al., 2017; De Lucia and Kühn, 2021), which the interface to high-level language R makes straightforward. This will allow a direct comparison between surrogate and POET's DHT as well as their composition. A further technical improvement will be the substitution of the R-driven PHREEQC coupling with the direct phreeqcRM module (Parkhurst and Wissmeier, 2015) or with a different and more efficient geochemical engine altogether, such as REAKTORO (Leal et al., 2020). A reimplementation of the flow sub-process, which is currently externally precalculated using the MUFITS simulator and therefore lacking feedback

between chemistry and hydrodynamics, is under consideration. Already available multiphysics packages such as MOOSE (Permann et al., 2020) or simulators such as OpenGeoSys (He et al., 2015) or TRANSE (Kempka, 2020) may be suitable to this end.

## 6   Conclusions

POET is a successful example of interdisciplinary cooperation. Bringing together competences from different scientific do-
mains makes it possible to achieve rapid and decisive technical progress and solve both practical and scientific problems more efficiently.

POET's master/worker architecture with non-contiguous grid partitioning represents a highly effective way to increase computational efficiency of parallel coupled simulations. Namely, it ensures optimal load balancing by scattering computational burden of cells invested by a reactive front across several work packages, and hence reducing the disparity of CPU-time re-
quired by the solution of each work package. The linear speedup observed in POET is only limited by the sequential part of the code, which can be further improved by a native C implementation avoiding the data transfer between the R runtime and the C/C++ domain.

Furthermore, we described an original implementation of fast DHT lookup to cache the results of geochemical sub-process simulations, enabling their reutilization in subsequent work packages and iterations. Our DHT implementation makes use of
fast one-sided MPI communication, in turn based on RDMA, and can be included as library into any application in which processes are expressed in terms of input and output tables, and hence not limited to geochemistry. Since by definition a DHT indexes only exact combinations of input variables, adjusting the rounding for each of these variables achieves an *approximated*





*lookup* with further speedup benefit at price of negligible accuracy loss. The optimal rounding for each variable, however, depends on the geochemical problem at hand, and it is hence left to the modeller to ensure adequate tuning of these parameters.

POET represents a foundation upon which it is possible to prototype and test novel algorithms, although currently with severe limitations regarding the implemented physics, most notably the lack of feedback between chemistry and hydrodynamics. However, POET allows to compute reactive transport models as large as SimDolKtz on single multi-cores machines in a realistic time, while compute clusters undoubtedly have no practical alternative for very large and long-term problems. The architecture of POET ensures that the same codebase efficiently runs on both systems, which is an invaluable practical benefit

in the geoscientific daily praxis. It is our opinion that only by reducing hardware requirements and computational costs with algorithmic improvements such as POET's DHT it is possible to perform the large scale, long-term complex reactive transport simulations and the associated risk and uncertainty analyses connected with subsurface utilization.

*Code availability.* POET is released under GPLv2 and can be downloaded from Zenodo (https://zenodo.org/record/4757913), DOI 10.5281/zenodo.4757913. Upon installation, the SimDol2D test case can be run out of the box.

**Appendix A:  Further Details on POET's DHT**

### A1   Application Programming Interface

POET's DHT library offers four operations: DHT_create, DHT_read, DHT_write and DHT_free. All information which is necessary for the management of the DHT is stored in a data structure called DHT-object shown in Listing 1.

**Listing 1.** The DHT struct declaration.

```
typedef struct {
        MPI_Win window;
        int data_size;
        int key_size;
        unsigned int table_size;
MPI_Comm communicator;
        int comm_size;
        uint64_t(*hash_func) (int, void*);
        void* recv_entry;
        void* send_entry;
void* mem_alloc;
        int read_misses;
        int evictions;
        unsigned int *index;
        unsigned int index_count;
DHT_stats *stats;
    } DHT;
```





Each worker has its own `DHT-object`. The first member of this structure is a handler for the local DHT part. Further, there are global DHT configuration parameters stored like the data and key size (line 3 and 4), i.e. only keys and values with that size or less can be stored in the DHT. This is due to the fact that the key-value pairs are saved directly in the hash table.

To determine the maximum of addressable buckets per process the table size per process is stored in `table_size`. The hash function in line 8 is used to hash the keys and is user defined. The struct also has pointers to intermediate buffers for the data that needs to be set or get (in line 9 and 10). These buffers are pre-allocated per task so there is no need to allocate memory after the initialization. `mem_alloc` in line 11 is the pointer to the allocated memory for the hash table. The members `index` and `index_count` are used for addressing the correct bucket and for collision resolution. Also there are counters for evictions,

read misses and a `DHT_stats` struct for additional evaluations. All variables given above are only used within the DHT library and do not need to be set or get by the user.

**Listing 2.** The `DHT_create` function signature.

```
DHT* DHT_create(
    MPI_Comm comm,
    unsigned int size_per_process,
    int data_size,
    int key_size,
    uint64_t(*hash_func)(int, void*));
```

To create a `DHT-object` the function `DHT_create` must be called. `DHT_create` requires information about the MPI

communicator to address the worker processes, the number of buckets per worker, and the data and key size in bytes. Furthermore, a pointer to a hash function must be passed.

When `DHT_create` is called, the required memory is allocated and a MPI_Window is created. This allows the execution of `MPI_Get` and `MPI_Put` operations for one-sided communication. Then the number of indices is calculated and, finally, all relevant data is entered into the `DHT-object` which is returned.

**Listing 3.** The `DHT_read` function signature.


```
int DHT_read(
    DHT *table,
    void* key,
    void* destination);
```

Data can be retrieved using the `DHT_read` function. This requires the created `DHT-object` and two pointers. One points to the input key and the other to an already allocated memory for the output data of the function. The data type of the stored elements is opaque for the DHT. Therefore, both pointers have to be casted to `void` before passing them to `DHT_read`.

At the beginning, the target process and all possible indices are determined. After that a `SHARED` lock on the address window for read access is done and the first entry is retrieved. Now the received key is compared with the key passed to the function.

If they coincide the correct data was found. If not it continues with the next index. If the last possible bucket is reached and the keys still do not match the read error counter is incremented. After the window has been released, the function returns with





a corresponding return value (read error or error-free read). The data to be read is written to the memory area of the passed pointer `destination`.

**Listing 4.** The `DHT_write` function signature.

```
int DHT_write(
          DHT *table,
          void* key,
          void* data);
```

`DHT_write` writes data into the DHT. The created `DHT-object` and pointers to the key and data value must be provided
to the function as input parameters. When `DHT_write` is called, the address window is locked with a `LOCK_EXCLUSIVE` for write access. Then the first bucket is read using `MPI_Get`. If the bucket is still empty or if the received key matches the passed key, the data is written using `MPI_Put`. If the bucket is occupied by a different (key,value) pair, the function continues with the next index until no more indices are available. In case all buckets are occupied, an eviction occurs and the last indexed bucket is replaced. After successful writing, the memory window is unlocked/released and the function returns.

**Listing 5.** The `DHT_free` function signature.

```
      int DHT_free(
          DHT *table,
          int* collision_counter,
          int* readerror_counter);
```

Finally, to free all resources after using the DHT, the function `DHT_free` must be used. This will free the `MPI_Window`, as well as the associated memory. Also all internal variables are released. The DHT handle must be passed here as input as well. Additionally, addresses for storing the values of the eviction and read error counters are passed.

## A2   DHT Dump and Restore

Besides the basic read or write command there is a possibility to save the current state of the DHT into a file. This is done by
using the function `DHT_to_file`. Thereupon all worker processes read out all written buckets of their local DHT memory area and write them into the specified file. Thus, after execution of the function a DHT blob is available, which can be re-read at any time with the function `DHT_from_file`. This is useful for checkpoint/restart, and it also allows the reuse of results from prior simulations for similar problems.

All read and write accesses to the file are performed via the MPI I/O interface to allow parallel and thus concurrent file
operations.

*Author contributions.* Conceptualization: MDL, MK and BS. Programming: AL, ML and MDL. Validation: ML, MDL and BS. Visualization: MDL, ML and AL. Performance Measurements: AL and ML. Writing: MDL, ML and BS with input from all co-authors. Revision: MK. Funding: MK and BS. All authors have read and agreed to the published version of the manuscript.





*Competing interests.* The authors declare no competing interests.

*Acknowledgements.* The authors want to thank Max Schroetter for identifying and solving an OpenMPI bug arising when running POET on a single node using shared memory (see bugreport https://github.com/open‑mpi/ompi/issues/8434).

M. De Lucia acknowledges the Helmholtz Association of German Research Centers - Initiative and Networking Fund for the funding in the framework of the project "Reduced Complexity Models" - reference number ZT-I-0010.



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
