# Peer review of "POET (v0.1): Speedup of Many-Cores Parallel Reactive Transport Simulations with Fast DHT-Lookups"

_Geoscientific Model Development, 2021_

## Referee Comment (RC2)

**Review of**
**POET (v0.1): Speedup of Many-Cores Parallel Reactive Transport Simulations with Fast DHT-Lookups**

The authors present a reactive transport simulator (POET) that makes use of HPC together with a Distributed Hash Table strategy to accelerate the simulations.

I find the paper well written, the literature review is extensive, the problem statements and results are clearly presented.

I just have minor change suggestions and recommendations to the authors.

- On subsection 2.4.3, (Approximated Lookup), the authors describe the approximate lookup strategy and give a complete remark about its advantages and disadvantages. The advantage is the fast retrieval of a previously computed equilibrium state for a given input vector. Because the given input vector is rounded to a certain number of digits (user-provided), the retrieved equilibrium state can be different than the actual fully computed one. If the equilibrium state is very sensitive with respect to some entry in the input vector, the resulting error can be substantial. I would like to point out that this disadvantage of their method can be eliminated (or greatly reduced) by adopting the first-order Taylor correction shown in Leal et al. (2020). This Taylor correction would also resolve the other disadvantage in their method in which the number of significant digits chosen for each input entry is problem-dependent. By using Reaktoro instead of PHREEQC, they would have access to the necessary derivatives to implement this correction. Their hash table would then not only save the input and output data, but also the sensitivity derivatives of the output vector with respect to the input vector. This reviewer is open to further discussion about this possibility with the authors and implementing existing PHREEQC features not yet available in Reaktoro.

- On Lines 73–80, I believe a more true account for the on-demand learning strategy presented in Leal et al. (2020) would be:

  *Leal et al. (2020) proposes an on-demand learning strategy for speeding up the chemical calculations without any need for training in advance. Their algorithm, implemented in Reaktoro (Leal et al. 2015), tries first to predict the result of an equilibrium calculation using a first-order Taylor extrapolation from a previously fully computed equilibrium state. If the prediction is not accurate enough, a new full equilibrium calculation is performed and the sensitivity matrix of the newly computed equilibrium state is saved for future predictions (this is the on-demand learning step). They use an on-demand clustering strategy to organize the learned computations so that an efficient search operation can be performed to find the most suitable learned equilibrium computation for the predictions based on chemical characteristics of the equilibrium states. The on-demand clustering strategy was devised to eliminate the curse of dimensionality that existed in their previous version of the algorithm (Leal et al. 2017), which relied on a nearest neighbor search strategy (see Kyas et al., 2020).*

  - Leal, A.M.M. (2015). Reaktoro: An open-source unified framework for modeling chemically reactive systems. www.reaktoro.org
  - Leal, A.M.M., Kulik, D.A., Saar, M.O. (2017). Ultra-Fast Reactive Transport Simulations When Chemical Reactions Meet Machine Learning: Chemical Equilibrium. http://arxiv.org/abs/1708.04825

- Is POET open-source? Is it available for download?

—

Allan Leal, ETH Zurich

---

## Author Comment (AC1)

**1 Reviewer 1**

We thank the anonymous reviewer for the positive comment on our work. We address the following concerns:

1. One shortage is that the largest run only occupies 719 core. The parallel size is too small to demonstrate the benefits this work can actually obtain in terms of large-scale simulation.

While this remark is certainly true from a computer science perspective, we point out the fact that computational resources larger than those explored in our work are rarely available to the geoscientists, and most certainly not on a daily basis. We believe that the efficient parallel architecture and the benefit of caching in DHT, as demonstrated in POET, are decisive advantages for the domain scientists: a relevant speedup for middle-sized simulations on limited hardware resources is arguably more valuable than very large scale simulations only tractable with large HPC infrastructure. Reactive transport simulations of the magnitude of the SimDolKtz test case are only sparsely encountered in literature. In our opinion, the scale of the simulations and of the computational resources addressed in this work cover a realistic - if not already ambitious - target for the practitioners.

2. My other concern is the portability of this work. Can proposed methods be applicable to other manycore-based systems, such as CPU-GPU heterogeneous system?

Offloading the embarassing parallel geochemical subprocess simulations to specialized hardware such as GPUs would of course be beneficial. However, to our good knowledge there is no available implementation of geochemical solver for GPU to date, which makes these consideration rather speculative at this point. The work package assemblage devised in the paper would still be optimal for GPU computations. Regarding the caching of results in the DHT in a heterogeneous context, the trade-off between caching in RAM and actual computation on accelerated hardware remains to be assessed.

**2 Reviewer 2**

We thank Dr. Leal for his review and positive comments.

1. On subsection 2.4.3, (Approximated Lookup), the authors describe the approximate lookup strategy and give a complete remark about its advantages and disadvantages. The advantage is the fast retrieval of a previously computed equilibrium state for a given input vector. Because the given input vector is rounded to a certain number of digits (user-provided), the retrieved equilibrium state can be different than the actual fully computed one. If the equilibrium state is very sensitive with respect to some entry in the input vector, the resulting error can be substantial. I would like to point out that this disadvantage of their method can be eliminated (or greatly reduced) by adopting the first-order Taylor correction shown in Leal et al. (2020). This Taylor correction would also resolve the other disadvantage in their method in which the number of significant digits chosen for each input entry is problem-dependent. By using Reaktoro instead of PHREEQC, they would have access to the necessary derivatives to implement this correction. Their hash table would then not only save the input and output data, but also the sensitivity derivatives of the output vector with respect to the input vector. This reviewer is open to further discussion about this possibility with the authors and implementing existing PHREEQC features not yet available in Reaktoro.

We agree that a first-order Taylor correction as described in Leal et al. (2020) would be a promising extension to the caching of computed solutions in Distributed Hash Tables. This would increase the size of the table since the sensitivities need to be stored as well, but as trade-off far less entries in the table would be required. This possibility is now mentioned in the conclusion:

"A promising extension to the DHT caching would be the application of the first-order Taylor correction as implemented in REAKTORO (Leal et al., 2020). This would potentially minimize the error introduced by approximated lookup while presenting a trade-off between the number of variables stored into the table (the sensitivities must also be stored), but with less overall and much more frequently reused entries."

2. On Lines 73-80, I believe a more true account for the on-demand learning strategy presented in Leal et al. (2020) would be: Leal et al. (2020) proposes an on-demand learning strategy for speeding up the chemical calculations without any need for training in advance. Their algorithm, implemented in Reaktoro (Leal et al. 2015), tries first to predict the result of an equilibrium calculation using a first-order Taylor extrapolation from a previously fully computed equilibrium state. If the prediction is not accurate enough, a new full equilibrium calculation is performed and the sensitivity matrix of the newly computed equilibrium state is saved for future predictions (this is the on-demand learning step). They use an on-demand clustering strategy to organize the learned computations so that an efficient search operation can be performed to find the most suitable learned equilibrium computation for the predictions based on chemical characteristics of the equilibrium states. The on-demand clustering strategy was devised to eliminate the curse of dimensionality that existed in their previous version of the algorithm (Leal et al. 2017), which relied on a nearest neighbor search strategy (see Kyas et al., 2020).

Thank you for this suggestion, which has been inserted into the manuscript with small modifications (including citations).

**3. Is POET open-source? Is it available for download?**

Yes, as pointed out in the "Code Availability" section, POET is released under GPLv2 and the source code can be downloaded from Zenodo (https://zenodo.org/record/4757913).